# The Mitochondrial Prohibitin (PHB) Complex in *C. elegans* Metabolism and Ageing Regulation

**DOI:** 10.3390/metabo11090636

**Published:** 2021-09-17

**Authors:** Artur B. Lourenço, Marta Artal-Sanz

**Affiliations:** 1Andalusian Centre for Developmental Biology (CABD), CSIC-Universidad Pablo de Olavide-Junta de Andalucía, Carretera de Utrera Km 1, 41013 Seville, Spain; 2Department of Molecular Biology and Biochemical Engineering, Universidad Pablo de Olavide, Carretera de Utrera Km 1, 41013 Seville, Spain

**Keywords:** mitochondrial prohibitin complex, metabolism, ageing

## Abstract

The mitochondrial prohibitin (PHB) complex, composed of PHB-1 and PHB-2, is an evolutionarily conserved context-dependent modulator of longevity. This extremely intriguing phenotype has been linked to alterations in mitochondrial function and lipid metabolism. The true biochemical function of the mitochondrial PHB complex remains elusive, but it has been shown to affect membrane lipid composition. Recent work, using large-scale biochemical approaches, has highlighted a broad effect of PHB on the *C. elegans* metabolic network. Collectively, the biochemical data support the notion that PHB modulates, at least partially, worm longevity through the moderation of fat utilisation and energy production via the mitochondrial respiratory chain. Herein, we review, in a systematic manner, recent biochemical insights into the impact of PHB on the *C. elegans* metabolome.

## 1. Introduction

Prohibitin proteins are strongly conserved from yeast to humans [1] and are related to bacterial HflKC, with which they share functional similarities [2]. In eukaryotes, prohibitins form a large multimeric complex called the mitochondrial prohibitin (PHB) complex [1,3,4]. The PHB complex is composed of 12–16 PHB-1/PHB-2 heterodimers assembled in a ring-shaped-like structure in the inner mitochondrial membrane facing the intermembrane space (reviewed in [5]). Both PHB-1/-2 subunits are ubiquitously and abundantly expressed, and are interdependent for protein complex formation, as the absence of either one of them leads to the absence of the full PHB complex [1,3,6]. Despite decades of work by many laboratories, the molecular function of the PHB complex is far from being clarified. It has been proposed to function as a chaperone-like protein that holds and stabilises mitochondrial proteins (reviewed in [1]) and as a lipid scaffold-like protein [7,8]. Although more work is still needed to better clarify its exact molecular function, evidence is accumulating for a direct impact of the PHB complex on mitochondrial functionality. In the yeast *Saccharomyces cerevisiae*, knockdown of the PHB complex leads to defects in mitochondrial membrane potential and to changes in mitochondrial morphology without an observable growth phenotype [6,9]. In multicellular eukaryotes, however, the PHB complex is essential for survival [3,7]. In mice, the post-natal tissue-specific absence of PHB in neurons results in neurodegeneration [10], and in β cells, PHB ablation impairs metabolic mitochondrial function and glucose homeostasis, leading to severe diabetes [11]. Furthermore, loss of PHB in podocytes results in kidney failure [12], and ablation in Schwann cells causes demyelinating neuropathy [13]. The mitochondrial phenotypes observed in these tissues and in mouse embryonic fibroblasts (MEFs), upon PHB ablation, include altered mitochondrial morphology, distribution, and dynamics, as well as aberrant cristae morphology [7,10,11,12,13]. In the nematode *C. elegans*, the postembryonic RNAi depletion of the PHB complex affects mitochondrial ultrastructure in muscle cells [3]. Moreover, worm tissues that rely heavily on mitochondrial function are more susceptible to PHB loss [3,14]. For example, the PHB complex is essential for somatic and germline differentiation in the larval gonad, resulting in its depletion in decreased fertility or even complete sterility. In addition, postembryonic PHB depletion leads to developmental delay, reduced body size, and slowed pharyngeal pumping and defecation [3]. PHB proteins play an important role in mitochondrial quality control, and their depletion induces the mitochondrial unfolded protein response (UPR^mt^), a stress response mechanism that reduces mitochondrial stress [15]. Likewise, PHB senses mitochondrial stress, and treatments that induce the UPR^mt^ increase PHB protein levels [1,3,16,17]. Importantly, PHB complex deregulation and mitochondrial dysfunction have been associated in different systems with physiological processes such as cancer, degenerative disorders, obesity, and ageing [18,19,20,21].

Ageing is a multifactorial process characterised by a progressive loss of functionality at the organelle, cellular, tissue, and organ levels that consequently will have an impact on the whole organism leading ultimately to death [22,23]. In recent decades, there has been a consistent trend of an increase in the world population age [24,25]. Massive changes in the age stratification structure of a country’s population necessary lead to economic, social, and health challenges [26,27]. Model organisms have been instrumental to shed light on the complex process of ageing and on age-related pathologies, leading to the identification of several conserved molecular pathways regulating ageing [23]. The insulin/insulin growth factor 1 (IGF-1) signalling pathway (IIS) is well conserved among species and a universal longevity regulator that was first identified to modulate ageing in *C. elegans* [28,29,30]. The IIS is activated by the binding of insulin-like peptides to its receptor, encoded by *daf-2* in *C. elegans*. DAF-2 activates AGE-1, and the downstream kinases AKT-1, AKT-2 and SGK-1 [31,32,33,34]. Activation of these kinases results in the phosphorylation of the transcription factor DAF-16 which is retained in the cytoplasm [35,36,37]. Upon inhibition of the IIS cascade, DAF-16 is activated [37,38,39] and triggers the expression of a plethora of genes involved in the regulation of lifespan [40]. The nematode *C. elegans* is one of the premier model organisms on ageing research due to its short life cycle and lifespan, its small size and transparency, the ease of its laboratory maintenance, and genetic manipulation. Moreover, fundamental eukaryotic cell biology and biochemistry processes are largely conserved between *C. elegans* and humans, including a wide spectrum of metabolic genes covering core metabolic pathways [41,42,43,44,45]. In fact, this is of particular relevance considering that metabolic alterations associated with nutrient-sensing pathways and mitochondria are a hallmark of ageing [46,47].

Prohibitins regulate replicative life span in yeast [9], senescence in mammalian fibroblasts [48], and promote longevity in worms [49]. Several years ago, we postulated that the PHB complex influences longevity through its effects on mitochondrial metabolism [5]. Later, PHB was found to modulate ageing in a metabolic-state dependent-manner [49,50,51]. Specifically, knockdown of the PHB complex shortens the lifespan of otherwise wild-type worms, while it markedly extends the lifespan of a large variety of *C. elegans* mutants. These include transforming growth factor-beta (TGF-β) signalling mutants, mutants with altered fat metabolism, mitochondrial electron transport chain (ETC) mutants, dietary restricted animals, and the long-lived IIS receptor *daf-2* mutants. Although biochemical data were largely lacking, the impact of PHB on longevity was suggested to be through the modulation of fat metabolism [49]. Since then, different analyses based on ^1^H NMR spectroscopy, liquid chromatography coupled with mass spectrometry (LC/MS), gas chromatography coupled with flame-ionisation detection (GC/FID), high-performance liquid chromatography (HPLC), and thin-layer chromatography (TLC) added extremely valuable molecular insights into the metabolic changes occurring in *C. elegans* upon PHB depletion in wild-type and IIS *daf-2* mutants [52,53]. Herein, we review these findings to provide a concise and systematic overview of the more recent biochemical insights into the effect of PHB on ageing regulation while setting the foundations for future studies.

## 2. The PHB Complex and Lipid Metabolism

In *C. elegans*, the PHB complex modulates fat content, as assessed by different fixed and live staining methods [49]. However, the complexity of the lipidome [54,55] is far from being captured through fat visualisation using dyes, as each of them has its own limitations [42,56]. In *C. elegans*, up to 35% of the dry body mass is composed of lipids, including free fatty acids, phospholipids, and triglycerides [54,57,58]. Remarkably, PHB deficiency has a wide impact on the *C. elegans* lipidome [52,53].

### 2.1. PHB Modulates the Whole-Worm Fatty Acid Composition

PHB depletion alters fatty acid (FA) composition, as assessed by GC/FID analysis. The effect of PHB depletion on FA composition is visible in developing L4 larvae. However, at the young adult (YA) stage, these changes become more noticeable, suggesting a more pronounced effect during adulthood or the accumulated consequence of altered FA metabolism. Overall, there is a trend towards an increase in shorter and monounsaturated FAs with a concomitant decrease in larger and polyunsaturated FAs upon PHB depletion. Surprisingly, while FA composition is clearly different between wild-type and *daf-2* mutants, the effect of PHB depletion on FA composition follows the same general trend in both genetic backgrounds—namely, an increase in the content of palmitoleic acid (C16:1), and a decrease of eicosapentaenoic acid (C20:5n3) and dihomo-γ-linolenic acid (C20:3n6) [53]. Previously, it was reported that the FA chain length and susceptibility to oxidation decreases sharply in long-lived mutants of the IIS pathway, correlating extremely well with the increased lifespan of these worms [59]. The lifespan increase that occurs in *daf-2* and in PHB-depleted *daf-2* mutants is accompanied by changes in the FA composition that overall follow this trend. However, the FA composition of PHB-depleted wild-type worms, which are short-lived, is altered in a similar fashion. Therefore, the whole-worm FA composition at the YA stage cannot, on its own, account for the effect of PHB on the worm’s longevity [53]. One possible explanation is that large changes in FA composition, which follow the same trend, mask other relevant changes in lifespan determination. Alternatively, the trend observed, although the same, might result from different contributions. For example, alterations in different cellular compartments and/or different lipid classes. In particular, alterations in lipids or lipid-related molecules implicated in ageing regulation, such as triglycerides (TAGs) or ascarosides [60,61,62]. Although purely speculative, the observed changes in FA metabolism could have an impact on ascaroside metabolism and, in this way, modulate ageing.

### 2.2. Sphingolipids and Glycerophospholipids Respond to PHB Depletion in a Genetic Background Dependent-Manner

An LC/MS analysis of whole worms at the YA stage identified many lipid species— namely, sphingolipids, such as sphingomyelin (SM) and ceramide (CER), and glycerophospholipids, such as phosphatidylcholine (PC) and phosphatidylethanolamine (PE), with an altered content in response to PHB deficiency and/or in response to *daf-2* mutation [53]. Specifically, both the SM pool, which, in other organisms, is mainly localised to the outer leaflet of plasma membranes [54], and the CER pool, important as a structural membrane lipid and required for surveillance of mitochondrial function [63], decrease their abundance upon PHB depletion. Similarly, PC and PE pools decreased in response to PHB depletion. Curiously, while *daf-2* also reduces the SM, CER, PC, and PE pools, PHB depletion in *daf-2* mutants only perturbs the CER pool with a further decrease. Of relevance, young *daf-2* mutants have much lower levels of PC + PE pool, compared to matching wild-types [53]. A complementary HPLC analysis of ageing worms showed that the glycerophospholipid pool of PC and PE, the two most abundant membrane lipids [64,65], is largely unaffected in PHB-depleted *daf-2* mutants, while it decreases markedly in PHB-depleted animals [53]. The differential effects identified on a whole worm basis, both at the YA stage and during ageing, could reflect a differential impact on the lipid composition of different organelles. The lipid composition of mitochondrial membranes is essential for the proper structure and function of the organelles. Membrane lipid biosynthesis occurs in an intimate interaction between the endoplasmic reticulum (ER) and mitochondria, as well as between mitochondrial membranes (Figure 1) [66]. In yeast, PHB genetically interacts with genes modulating mitochondrial phospholipid biosynthesis, in particular cardiolipin (CL) and PE, affecting the distribution of CL and PE by clustering them at distinct sites of the internal mitochondrial membrane [8,67,68]. Additionally, in MEFs, PHB cooperates with the mitochondrial cochaperone DNAJC19, for which DNJ-21 is the worm homolog, in the remodelling of mitochondrial membrane phospholipids. Specifically, lack of PHB complexes alters CL acylation, while the transcriptional response of PHB deficient cells shows altered lipid metabolism, most prominently cholesterol [69]. The PHB complex has been, therefore, suggested to act as a membrane organiser affecting the distribution of mitochondrial membrane lipids [8,67,68,69].

### 2.3. PHB Depletion Strongly Alters the Triacylglycerides Pool

In worms, the PHB complex alters glycerolipids at the YA stage. As assessed by whole worm LC/MS analysis, and similar to *daf-2* mutants [70,71], PHB depletion increases the content of the large majority of glycerolipids species, while further increasing diacylglyceride (DAG) and TAG pools in *daf-2* mutants. Strikingly, PHB depletion increases, mostly, TAG species with a longer average chain length, while in *daf-2* mutants, lack of PHB mostly increases TAG species with a shorter average chain length. Additionally, while PHB depletion increases TAGs irrespective of the average degree of unsaturation, the effect of knocking down PHB in *daf-2* mutants is restricted to TAGs with a low average degree of unsaturation. The FA composition of the TAG pool of whole worms, separated by TLC, indicates that PHB-depleted animals have a reduced content of shorter monounsaturated FAs than PHB-depleted *daf-2* mutants, which concomitantly have a higher content of longer saturated FAs. Opposite to the glycerophospholipid pool, the TAG pool in PHB-depleted animals increases during ageing, while in PHB-depleted *daf-2* mutants, it is much less affected [53]. The TAG pool is modulated by PHB in an IIS-dependent manner, which suggests a differential balancing and mobilisation of the TAG pool during ageing (Figure 1).

### 2.4. PHB Deficiency Affects Different Lipid-Related Organelles

Lipid homeostasis within a cell is achieved through the dynamic interaction between different organelles such as mitochondrion, ER, lipid droplet (LD), and yolk particle (YP) [72,73,74,75,76]. Mitochondria are particularly tightly connected with the ER but also with LDs [72,75,76]. In the worm’s intestine, neutral fat is accumulated in LDs, ubiquitous fat storage organelles, which are then mobilised according to the organism’s needs for membrane synthesis and energy [77,78]. Interestingly, in young animals PHB and DAF-2 affect LDs homeostasis differently, compared to wild-type animals, *daf-2* mutants have higher LD intestinal coverage, whereas there is a much weaker effect in PHB-depleted animals. Strikingly, PHB depletion in *daf-2* mutants synergistically increases the LD intestinal coverage of larger LDs [53]. Consistently, protein levels of ATGL-1, the worm homolog of the mammalian rate-limiting lipolytic enzyme ATGL and required for *daf-2* longevity [79], are higher in wild-type worms than in *daf-2* mutants during ageing. Moreover, while PHB depletion does not affect ATGL-1 levels in wild-type animals, it consistently further lowers ATGL-1 levels in *daf-2* mutants. In parallel, PHB depletion differentially deregulates yolk homeostasis in wild-type and in *daf-2* mutants [53]. YPs, produced through vitellogenesis, carry lipids such as TAGs and PLs to the gonad, where they are taken up by developing oocytes. Particularly during the reproductive period, vitellogenesis has a major impact on lipid homeostasis [80,81,82,83,84,85]. The PHB complex is essential for germline function, and its depletion leads to sterility [3]. Maybe as a consequence, PHB depletion accumulates large amounts of displaced yolk through the worm body during ageing. Strikingly, PHB-induced yolk accumulation is suppressed by *daf-2* in aged worms (Figure 1). The ER is involved in the formation of LDs [86,87] and of YPs [88]. Mitochondrial contacts with the rough ER are important for lipoprotein secretion and systemic lipid homeostasis [88,89]. Importantly, PHB genetically interacts with genes involved in mitochondria-ER contact sites [90]. Interestingly, PHB depletion disrupts ER homeostasis, as assessed by a UPR^ER^ stress reporter, suggesting deregulation in the interaction between mitochondria and ER [53]. A recent publication shows that mitochondrial dysfunction caused by PHB deficiency leads to ER stress in Schwann cells of conditional knockout mice [13]. Curiously, *daf-2* mutants are protected against ER stress, which has been linked to its longevity phenotype [91,92]. Indeed, while PHB depletion induces ER stress in otherwise wild-type worms, *daf-2* mutant animals are protected, providing a plausible link between the ER and the PHB complex in ageing determination (Figure 1) [53].

## 3. PHB Has a Broad Impact on the *C. elegans* Metabolome

The metabolic effect of PHB depletion is not restricted to fat but has a much broader effect on the metabolic network [52]. Indeed, the ^1^H NMR metabolic profiles of whole worm extracts show the same general patterns evidenced by lipid analyses—namely, the metabolic changes due to PHB depletion become more pronounced throughout development. Strikingly, the metabolic profiles at the YA stage, reveal that PHB depletion has a stronger effect in wild-type animals than in *daf-2* mutants. Specifically, the ^1^H NMR metabolic profiles uncover changes in carbohydrate and amino acid metabolism (Figure 1) [52].

### 3.1. PHB Adjusts the Content of Essential and Non-Essential Amino Acids

PHB depletion perturbs the abundance of a large spectrum of amino acids at the L4 and YA stages. At the YA stage, both, PHB deficiency and *daf-2* mutation alter the content of amino acids, leading to PHB depletion in *daf-2* mutants to a further readjustment of amino acid metabolism [52]. It has been described that supplementation of different amino acids can modulate *C. elegans* lifespan [2]. Interestingly, PHB depletion, specifically in otherwise wild-type animals, decreases the content of two branched-chain amino acids, leucine and valine, to the levels of *daf-2* mutants, in which these changes are entirely DAF-16 dependent [93]. Although mild, wild-type worms have a higher content of alanine than *daf-2* mutants. Strikingly, upon PHB depletion, alanine content is further increased in wild-type animals and further reduced in *daf-2* mutants. Importantly, an inverse correlation between alanine levels and yeast chronological lifespan has been reported [94]. Similar to its effect on wild-type worms, PHB depletion in *daf-2* mutants decreases the content of glutamate, while it increases the content of glutamine. Among other metabolic pathways, glutamate/glutamine metabolism is important for replenishing the tricarboxylic acid (TCA) cycle through their oxidative deamination [95]. Curiously, glutamate/glutamine metabolism has been described to be adjusted in impaired mitochondrial mutants with an altered lifespan [96]. The widespread impact of the PHB complex on amino acid metabolism (Figure 1), both in nutritionally essential and non-essential amino acids [97], reinforces the idea of a broad reorganisation of the metabolic network [52].

### 3.2. PHB Deficiency Rewires Carbohydrate and Energy Metabolism

Mitochondria are essential organelles in energy metabolism carrying out the TCA cycle and oxidative phosphorylation (OXPHOS) [98,99]. In mice, knockdown of liver PHB leads to an adjustment of whole-body energy homeostasis [100]. In aged worms, knockdown of PHB selectively increases oxygen consumption in *daf-2*, indicating that these worms sustain higher mitochondrial function in a long term [49]. PHB-depleted worms have a lower content of the TCA metabolite succinate, compared to wild-type animals, in line with the changes in TCA-related amino acids. Curiously, the amount of succinate is also reduced in *daf-2* mutants but unaltered upon PHB depletion, which suggests that PHB adjusts the TCA cycle in wild-type animals but not in *daf-2* mutants [52]. *daf-2* mutants shift metabolism away from the TCA cycle towards the glyoxylate cycle [93,101,102]. The glyoxylate cycle is a variation of the TCA cycle that bypasses the decarboxylation steps and, among other things, enables the interconversion of fats and carbohydrates [103]. In *C. elegans*, endogenous glucose is stored in the form of trehalose, a non-reducing disaccharide, or glycogen, a branched polysaccharide [97]. Trehalose presumably acts as a stress protectant against multiple stresses [104,105,106,107,108,109] and as a longevity assurance sugar in *C. elegans* [107,110,111]. Although not to the same extent as *daf-2* mutants, PHB depletion causes the accumulation of large amounts of trehalose. Interestingly, depleting PHB in a *daf-2* mutant has an additive effect in trehalose accumulation [52]. Under a high sugar diet, shifting sugar storage from glycogen to trehalose promotes lifespan and health span in a DAF-16/FOXO-dependent manner. Specifically, lifespan increases as a result of high levels of internal trehalose through autophagy upregulation [111]. Moreover, trehalose supplementation from the YA stage significantly extends lifespan [107]. However, the requirement of trehalose for *daf-2* longevity has been recently brought into question, because a deficiency in trehalose synthesis mildly shortened *daf-2* lifespan [109], compared with previous studies [107,111]. Similar to trehalose supplementation, exogenous addition of lactate, which induces mitohormesis, leads to stress resistance and survival [112]. Lactate is an important bioenergetic metabolite formed either from fermentation or through aerobic glycolysis. Wild-type animals at the YA stage present larger amounts of lactate than *daf-2* mutants. Moreover, similar to the effect on the pyruvate-related amino acid alanine, PHB depletion triggers an opposite effect on the content of lactate. While in otherwise wild-type animals, PHB depletion showed a trend to increase the content of lactate, in *daf-2* mutants, PHB depletion showed a trend to decrease lactate content. This adjustment, as well as that of the pyruvate-related amino acid alanine, suggests a tuning of fermentative metabolism with possible implications in energy balance and longevity [52].

## 4. PHB-Mediated Ageing Regulation in a Metabolic Perspective

Mitochondrial function and metabolic homeostasis, crucial in ageing regulation [113], is severely disturbed by PHB depletion in a metabolic-state-dependent manner. Indeed, PHB depletion shortens the lifespan of wild-type animals, while enhancing the longevity of a plethora of metabolically compromised mutants, including targets of rapamycin complex 2 mutants *sgk-1* and *rict-1*, and of the IIS receptor *daf-2* mutants [49,50,114], linking PHB functions in mitochondria with cellular metabolism [2,31,52]. PHB deficiency also extends the lifespan of both, *nhr-49* and *fat-7* mutants [49]. NHR-49 is a key regulator of fat mobilisation, modulating fat consumption and maintaining a normal balance of FA saturation, while FAT-7 is required for the synthesis of monounsaturated fatty acids [115]. Collectively, our data suggest that the excess of TAGs accumulated in PHB-depleted *daf-2* mutants is accommodated in larger intestinal LDs, which can be mobilised during ageing through tighter regulation of ATGL-1 (Figure 1) [53]. Cardiac fatty acid oxidation (FAO) is impaired in short-lived PHB2 cardiac-specific knockout mice through downregulation of carnitine palmitoyltransferase, a rate-limiting enzyme in mitochondrial FAO, [116]. Moreover, PHB deficient Schwann cells show reduced biosynthesis of fatty acids [13]. It would be interesting to investigate if these mechanisms are conserved in the nematode and whether they might be differentially regulated in *daf-2* mutants, which could explain the differential effect of PHB in gut LDs. Importantly, yolk production, at the expense of the gut by a process of general autophagy, has been proposed as a major driver of worm senescence [117]. Lipotoxicity through ectopic yolk accumulation has been suggested to contribute to the opposing PHB ageing phenotype because PHB-induced yolk accumulation is suppressed by *daf-2* in aged worms [53]. Interestingly, the specific depletion of PHB in mice hepatocytes causes a dramatic imbalance of lipid storage from adipose tissue to the liver [100]. We propose that PHB modulates yolk accumulation and TAG storage lipids in an IIS dependent-manner (Figure 1), reinforcing the connection between PHB and lipid metabolism in ageing regulation. Previously, PHB deficiency was shown to shorten the lifespan of *aak-2* [49]. The AMP-dependent kinase AAK-2 has been implicated in coupling energy levels with signals from the IIS pathway to modulate lifespan in *C. elegans* [118,119]. The PHB complex is also important for mitochondrial DNA maintenance [120] and stabilisation of respiratory complexes and supercomplexes [121,122]. Strikingly, different OXPHOS mutants (*gas-1*, *mev-1*, and *isp-1*) also extend their lifespan upon PHB depletion [49]. The *gas-1*, *mev-1*, *isp-1* genes encode subunits of the mitochondrial electron transport chain complexes I, II, and III, respectively, emphasising the importance of OXPHOS and PHB in lifespan regulation [53].

A key determinant of structural and functional integrity of eukaryotic membrane-bound organelles is membrane lipid composition [123,124]. As mentioned above, the PHB complex is important for mitochondrial membrane maintenance. Interestingly, depletion of DNJ-21, which physically and functionally interacts with PHB in remodelling mitochondrial cardiolipin, mimicked the differential effect of PHB deficiency on the lifespan of wild type and *daf-2* mutants [53]. It would be interesting to investigate if DNJ-21 phenocopies other PHB phenotypes including effects on LD and YP, and lifespan extension in other metabolically compromised mutants. Importantly, reduced sphingolipid and ceramide synthesis extend the lifespan of worms and flies [125]. Recently, PHB was found to be required for the lifespan extension conferred by reduced sphingolipid synthesis in *C. elegans*. Moreover, in the same study, PHB depletion was found to suppress the impaired mitochondrial homeostasis, lipogenesis, and yolk formation of *sgk-1* [114]. In summary, the PHB complex plays a crucial role in the determination of mitochondrial function by affecting, directly or indirectly, the maintenance and composition of mitochondrial membranes and by leading to systemic changes in metabolism, which ultimately, differentially modulate lifespan depending on the metabolic state of the animals (Figure 1).

## 5. Future Perspectives

Tremendous efforts have been made thus far to better characterise the effect of the PHB complex on the *C. elegans* metabolic network. Still, to give a more refined perspective on the broad metabolic alterations modulated by the PHB complex other levels of global data, such as RNA sequencing or proteomics data, would be extremely valuable. In the near future, it would be very interesting to assess the real relevance of several aspects raised so far to the PHB-mediated regulation of ageing—namely, the perturbation of mitochondrial membrane composition, the deregulation of yolk accumulation, the tight mobilisation of fat stores, and the balance of sugars storage. Specifically, it would be relevant to assess the role of yolk/lipoproteins on lifespan regulation and whether it is through vitellogenesis and/or ectopic yolk accumulation. Similarly, in the context of the positive impact of PHB depletion in the lifespan of *daf-2* mutants, it would be interesting to determine whether increased availability of storage lipids and sugars and/or a more efficient mobilisation of these energy stores are relevant for lifespan regulation. Moreover, investigating the implication of peroxisomes would be relevant considering their implication on FAO and mitochondrial function in other life-extending paradigms [126]. Herein, we highlight some lines of research we are currently pursuing aimed at understanding the intriguing ageing phenotype of the PHB complex which is relevant to understand human ageing.

## Figures and Tables

**Figure 1 metabolites-11-00636-f001:**
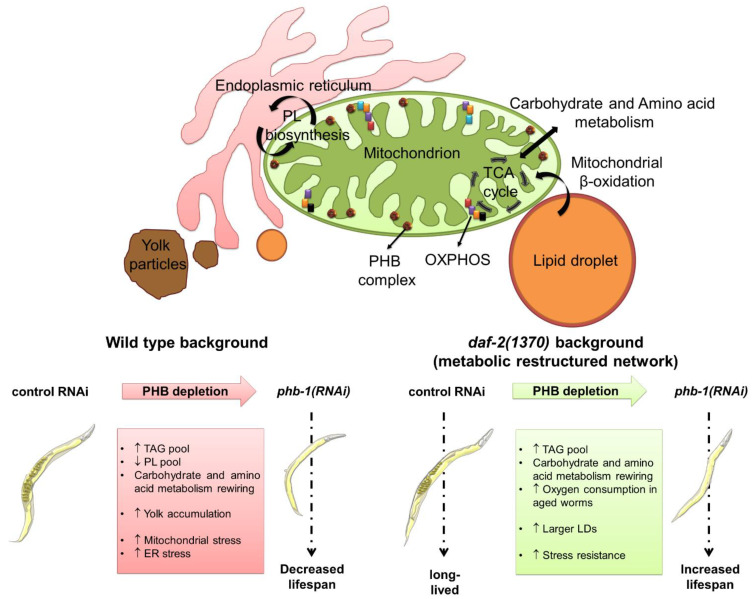
Metabolic rewiring in PHB-mediated ageing regulation. The PHB complex modulates mitochondrial membrane lipid composition, presumably by altering the balance of PL, leading to deregulation of mitochondrial function, including oxidative phosphorylation (OXPHOS). Mitochondrial dysfunction affects amino acid and carbohydrate metabolism in an IIS dependent-manner. Concomitantly, mitochondrial dysfunction reverberates, in an IIS-dependent manner, in other organelles such as yolk, lipid droplets, and the endoplasmic reticulum, leading to opposing ageing phenotypes.

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
