# Peer review of "The Mitochondrial Prohibitin (PHB) Complex in C. elegans Metabolism and Ageing Regulation"

_metabolites, 2021, doi:10.3390/metabo11090636_

Round 1
Reviewer 1 Report
This review gives an overview of prohibitin function in C. elegans and its relation to metabolism and aging. It is written by the leading research group on this specific topic. The authors focus on the description of several omics approaches, carried out in their research group, that may hint at the function of the prohibitin complex in C. elegans.
I have a few major concerns about this review.
First, concerning the available literature, the topic of this review is very narrow. This is why most of the reviewed material is based on essentially three publications of this research group (references 40, 43, and 44 which were cited in the review 11, 12, and 15 times, respectively). In a review, I would expect a somewhat broader topic that can cover a wider range of studies.
Second, the general structure of the review is good but it lacks balance in the introduction section. Disproportional attention is given to societal aspects of ageing as well as the IIS pathway. More focus on prohibitins in the introduction would be more appropriate for this review.
Third, this review requires editing by a native speaker because many style, grammatical and spelling errors can be found in this manuscript. For example, the authors should avoid the overuse of introductory adverbs. In some sections, most sentences start this way - specifically (7x), interestingly (7x), importantly (9x), strikingly (6x), curiously (8x)…. Moderation of these intensifiers is required. Repetition of specific (often unclear) expressions should be avoided too (‘in a metabolic-state dependent manner’ – see below, ‘PHB depletion in otherwise wild-type animals’, etc.)
Specific remarks:
Line 17-18: unclear, rephrase.
Line 62: C. elegans’ high degree of homology with humans is somewhat exaggerated. It would be safer to state that basic eukaryotic cell biology and biochemistry are largely conserved between C. elegans and humans.
Line 80: “The PHB complex modulates fat content in a metabolic status dependent manner…”. This is a confusing description that occurs many times throughout the text. It does not exactly refer to any specific metabolic status and is a very vague description. Essentially, I assume that the authors refer to wild-type worms versus daf-2 mutants and that PHB knockdown results in different phenotypes (lifespan, fat content, …) in those two genetic backgrounds. Referring to daf-2 exclusively as ‘metabolic status dependent’ may be dangerous in that sense: the daf-2 metabolism is indeed completely restructured, but also other aspects of its cell physiology such as redox balance and expression of chaperones are altered and may also influence mitochondrial function and the resultant effect of PHB knockdown. It would be more correct to refer to the daf-2 mutation or reduced IIS rather than using the cryptic ‘metabolic status dependency’.
Line 88-94: Lacks flow. Some sentences about FA composition during development and adulthood are followed by the effect of PHB knockdown on FA composition. Nothing is mentioned on why a description of FA composition is developing worms is important in this context.
Line 93: “PHB depletion in otherwise wild-type animals”. This is an unnecessary long description that occurs on several occasions in the manuscript. ‘PHB-depleted animals’ or ‘phb(RNAi) worms’ refers to the same condition. Where no wild-type background is used, e.g. ‘PHB-depleted daf-2 mutants’ could be used.
Line 95: “…they respond to PHB inhibition with the same…”
Line 98-99: not clear. More explanation is needed about the correlation – what is meant with “correlates extremely well”? Include the mechanistic line of reasoning here. I assume the authors hint at the accumulation of oxidized FA as a lifespan shortening factor? Is there any literature supporting this?
Line 116: “A complementary HPLC analysis of aging worms showed…”
Line 117-120: The authors state that PC + PE levels decrease in PHB-depleted WT over age while these levels remain unaffected in the corresponding daf-2. Taking into account only the kinetics, this is correct, but a probably biologically more relevant aspect is that, in PHB knockdowns, PC + PE levels are much lower in young daf-2 worms compared to matching wild-types. In wild types, the PC/PE levels decrease with age, while daf-2 maintains its low levels.
Line 135: Figure 1 – in the red and green boxes: “Carbon and amino acid rewiring”. Correct? Or do the authors mean “Carbohydrate” instead of “Carbon”?
Line 174: “…, whereas there is no significant effect in PHB-depleted…”
Line 189: PHB depletion induces ER stress in WT, but not in daf-2 mutants. According to the authors, this links ER and PHB to ageing. Why? Maybe daf-2 worms, known to be hyper-resistant to most kinds of stress, do not suffer as much as WT from PHB knockdown and hence, do not need to activate their UPRer upon PHB depletion. This phenomenon may be completely independent from their longevity phenotype.
Lines 194-201: improve the logic/flow/clarity of the story. The authors provide facts about PHBs, vitellogenins, reproduction, and ageing, but do not link these in a logical train of thoughts that leads to a conclusion. It is not clear what is the point. The latter problem also occurs in some other sections.
Line 211: ‘diapause’ is not relevant here.
Line 221: Is it a readjustment? Or deregulation?
Line 236: ‘Altered lifespan’: decreased or increased? If both, is it relevant to ageing?
Line 237: delete ‘nutritionally’ twice.
Line 247: how does this sentence add to the story? Is there any connection with the previous or next sentence?
Line 254: “…, PHB depletion causes accumulation of large amounts…”
Line 254: “Curiously…” why is the additive effect of PHB depletion and daf-2 mutation on worm trehalose levels curious? Because they are assumed to act on trehalose via the same pathway?
Line 267-269: effects of PHB depletion on metabolites are described but not discussed. What is relevant here? Is there any explanation of the opposite lactate trends? Why is it mentioned?
Line 285-295: This part is all over the place and lacks a coherent storyline.
Line 296: replace ‘longevity’ with ‘lifespan’ (gas-1 and mev-1 mutant worms are short-lived under standard conditions).
Line 305: replace ‘ageing regulation’ with ‘lifespan’.
Line 307: I was a bit disappointed reading the “future perspective” section. Basically, it reads that we need more ‘omics’ data to elucidate the mysterious function of PHBs. I was hoping to find more targeted approaches based on current knowledge on the topic. Some of these are given in the second part of this section but could be more specific.
Reviewer 2 Report
A manuscript by Artur Lourenço and Marta Artal-Sanz reviews effects of phb-1 knockdown on longevity and biochemical phenotypes in C elegans. I have two main comments.
First, considering the complexity of the subject, the amount of available data, and sometimes apparent contradiction of the results on one hand and experience of the authors with this topic on the other, it would be helpful to include more analysis and discussion in the paper. As it stands, it looks more like a mere description of results and enumeration of published papers. An authors’ opinion along with synthesis of those results and a mechanistic hypothesis how PHB may function and produce observed phenotypes would help a non-expert to navigate the paper and the field.
Second, a significant portion of the review is devoted to the description of phb-1 fat phenotypes. While the authors mention various lipid classes and outline how their levels and/or distribution are affected by phb-1 in various genetic backgrounds, there’s no word on ascarosides. The latter molecules contain fatty acid side chains and one might expect those to be affected by perturbation in lipid metabolism and mitochondria function in general. Given many important roles that ascarosides play in worm life, those potential perturbations may have biological consequences. Are there any works on this subject? If not, it would be interesting to hear the authors’ opinion.
Finally, English grammar needs some editing.
Overall, I find the paper useful, interesting, and worth publishing, if the above comments can be addressed.
Round 2
Reviewer 1 Report
The manuscript was much improved by the authors.
I only have a few minor remarks left.
Line 87: “metabolically compromised mutants, including the long-lived IIS receptor daf-2 mutants”. I understand that the authors also refer to other mutants, including mitochondrial mutants, for which I agree that their metabolism is likely compromised, but I would be reluctant to define the daf-2 metabolism as being compromised. In these mutants, a very specific and elaborated metabolic program is triggered by DAF-16 activation.
Line 236-238: the second part of the sentence does not provide new information and essentially reiterates the first part.
Line 262-264: Very recently, the role of trehalose in daf-2 longevity has been refuted by Rasulova et al. (METABOLITES 11 (2). doi:10.3390/metabo11020105). This may be considered in this paragraph.
Line 314: “lifespan regulation” is a safer concept here than “ageing regulation” as it avoids the discussion whether or not the mitochondrial mutants gas-1 and mev-1 are short-lived because they actually age faster or whether they suffer a mitochondrial pathology (that can be rescued) which kills them early.
If the authors can take these remarks into account this manuscript can be accepted.
Author Response
Dear Referee,
We would like to thank you again for your time and effort in reviewing our manuscript, for your careful reading and constructive suggestions.
A point-by-point response to all the comments follows below (original comments are quoted in bold).
Reviewers' Comments to Author:
Reviewer 1:
The manuscript was much improved by the authors.
I only have a few minor remarks left.
We are happy to hear the reviewer is now satisfied. We have below responded and modified all the minor remarks raised by the reviewer.
Line 87: “metabolically compromised mutants, including the long-lived IIS receptor daf-2 mutants”. I understand that the authors also refer to other mutants, including mitochondrial mutants, for which I agree that their metabolism is likely compromised, but I would be reluctant to define the daf-2 metabolism as being compromised. In these mutants, a very specific and elaborated metabolic program is triggered by DAF-16 activation.
We agree with the reviewer and are thankful for this remark. We have now changed this sentence to: “Specifically, knockdown of the PHB complex shortens the lifespan of otherwise wild-type worms, while it markedly extends the lifespan of a large variety of C. elegans mutants. These include transforming growth factor beta (TGF-β) signalling mutants, mutants with altered fat metabolism, mitochondrial electron transport chain (ETC) mutants, dietary restricted animals and the long-lived IIS receptor daf-2 mutants”.
Line 236-238: the second part of the sentence does not provide new information and essentially reiterates the first part.
We thank the reviewer for pointing this out. We have now replaced the sentence to provide the information missing. Now the sentence is: “Strikingly, upon PHB depletion, alanine content is further increased in wild type animals and further reduced in daf-2 mutants.”
Line 262-264: Very recently, the role of trehalose in daf-2 longevity has been refuted by Rasulova et al. (METABOLITES 11 (2). doi:10.3390/metabo11020105). This may be considered in this paragraph.
We apologise for missing this relevant recent publication and thank the reviewer for bringing it to our attention. We have now included the following sentence: “However, the requirement of trehalose for daf-2 longevity has been recently brought into question, because deficiency in trehalose synthesis mildly shortened daf-2 lifespan {Rasulova, 2021 Metabolites} compared to previous studies {Honda, 2010 Aging Cell, Seo, 2018 PNAS}”. Now lines 304-307.
Line 314: “lifespan regulation” is a safer concept here than “ageing regulation” as it avoids the discussion whether or not the mitochondrial mutants gas-1 and mev-1 are short-lived because they actually age faster or whether they suffer a mitochondrial pathology (that can be rescued) which kills them early.
Changed. Thanks for the reviewer for this nice remark.
If the authors can take these remarks into account this manuscript can be accepted.
In closing, we would like to once more thank the Reviewers for the constructive and positive input. We appreciate all the feedback we get that helps us improving our paper. We do hope that you will find our revisions adequate for publication of our work now in Metabolites.